# Association between Reactogenicity and Immunogenicity in a Vaccinated Cohort with Two mRNA SARS-CoV-2 Vaccines at a High-Complexity Reference Hospital: A *Post Hoc* Analysis on Immunology Aspects of a Prospective Cohort Study

**DOI:** 10.3390/vaccines12060665

**Published:** 2024-06-17

**Authors:** Joaquín Sáez-Peñataro, Gonzalo Calvo, Juan Bascuas, Maria Mar Mosquera, Maria Ángeles Marcos, Natalia Egri, Ferran Torres

**Affiliations:** 1Medicines Division, Department of Clinical Pharmacology, Hospital Clínic de Barcelona, Institut d’Investigacions Biomèdiques August Pi i Sunyer (IDIBAPS), Universitat de Barcelona, 08036 Barcelona, Spain; gcalvo@clinic.cat (G.C.); bascuas@clinic.cat (J.B.); 2Microbiology Department, Hospital Clinic, Institute for Global Health, University of Barcelona, 08036 Barcelona, Spain; mdmosquera@clinic.cat (M.M.M.); mmarcos@clinic.cat (M.Á.M.); 3CIBERINF, 28029 Madrid, Spain; 4Immunology Department, Hospital Clínic de Barcelona, Institut d’Investigacions Biomèdiques August Pi i Sunyer (IDIBAPS), Universitat de Barcelona, 08007 Barcelona, Spain; egri@clinic.cat; 5Department of Biostatistics, Autonomous University of Barcelona, 08193 Barcelona, Spain; ferran.torres@uab.cat

**Keywords:** mRNA vaccines, reactogenicity, immunogenicity, pharmacovigilance, real-world data

## Abstract

Enhancing our comprehension of mRNA vaccines may facilitate the future design of novel vaccines aimed at augmenting immune protection while minimising reactogenic responses. Before this design is carried out, it is important to determine whether adaptive immunity correlates with the reactogenicity profile of vaccines. We studied a large cohort that was vaccinated with mRNA vaccines to answer this question. This was an observational study with real-world data. Reactogenicity data were obtained from the VigilVacCOVID study. Immunogenicity (humoral and cellular) data were retrieved from health records. One main population (n = 215) and two subpopulations were defined (subpopulation 1, n = 3563; subpopulation 2, n = 597). Sensitivity analyses were performed with subpopulations 1 and 2 to explore the consistency of results. We analysed the association of the intensity and types of adverse reactions with the development and quantity of elicited antibody titres. As an exploratory analysis in subpopulation 1, we assessed the association between reactogenicity and cellular immunogenicity. A higher incidence of fever, malaise, and myalgia including severe cases was significantly associated with the development and quantity of positive antibody titres. No significant findings were observed with cellular immunity. We observed a positive association between immunogenicity and reactogenicity. These findings can be relevant for the future development of our understanding of how mRNA vaccines function.

## 1. Introduction

The short-term safety and efficacy of SARS-CoV-2 mRNA vaccines have been widely characterised in pivotal clinical trials [1,2]. After carrying out pivotal clinical trials, several large scale “real-world” studies were conducted, reporting that SARS-CoV-2 vaccines were safe [3,4,5]. In these studies, the most frequently reported adverse reactions (ARs) were local (e.g., pain at the injection site) or systemic ARs (e.g., fever, myalgia, or malaise) [1,2]. These were usually self-limited and manageable ARs, which further reinforces the positive benefit–risk balance of the currently available SARS-CoV-2 vaccines.

However, given the emergence of several variants of concern, it is highly likely that new vaccine platforms that target new age strata, new antigens, and different booster doses will continue to be developed.

Undoubtedly, the ideal vaccine construct should effectively mobilize both the innate and adaptive immune responses without provoking systemic inflammation, which could potentially lead to serious ARs. One remaining question is whether vaccine-associated ARs can be uncoupled from the elicited immune response in an ideal vaccine construct [6,7]. Various authors have described binding and neutralizing antibodies as potential immune correlates of protection for SARS-CoV-2 vaccines [8,9]. The question that follows from their research is whether this adaptive immunity is, to some extent, correlated with the reactogenicity profile of vaccines in current vaccine constructs.

Recent studies have suggested that a higher reported frequency of ARs after vaccination could be associated, to some extent, with an increase in antibody titres [10,11]. Krammer et al. reported that individuals with pre-existing SARS-CoV-2 immunity showed a significant spike antibody immunogenicity and experienced more severe reactogenicity after the first dose compared to individuals without pre-existing antibodies [11]. Additionally, Debes et al. reported that spike IgG antibodies were higher in vaccinees with prior SARS-CoV-2 infection and those that reported clinically significant reactions after vaccination [12]. However, to date, the potential link between immunogenicity and reactogenicity has yielded contradictory results [13,14,15]. Therefore, the association between reactogenicity and immunogenicity after the first and second doses remains to be confirmed in larger vaccinated cohorts, especially in a setting that can be extrapolatable to clinical practice (i.e., in an heterogenous and more representative population beyond health-care professionals (HCPs)).

Overall, a deeper understanding of the immunological correlates of reactogenicity can assist in the future development of newer vaccine constructs through which immune protection can be boosted without hindering a proportional increase in reactogenicity. This becomes increasingly important given the expected increase in mRNA technology development in several clinical indications beyond SARS-CoV-2 vaccines [16].

In our prior VigilVacCOVID study, we observed a correlation between reactogenicity and immunogenicity [17]. This was evidenced by the consistent association of a higher incidence of AR with younger age and prior exposure to SARS-CoV-2, which is also in line with other published reports [18]. Conversely, reduced vaccine reactogenicity was linked to a compromised immune status, leading to improved tolerability. Following this rationale, we took advantage of the reactogenicity data collected within the framework of the study to analyse the immunogenicity data and substantiate the hypothesis of a potential link between reactogenicity and immunogenicity in a large, vaccinated cohort under the conditions of clinical practice.

The VigilvacCOVID study was an observational, single-centre post-authorisation study aimed at characterising short-term reactogenicity after vaccination with mRNA-1273 and BNT162b2 vaccines in two different populations: HCPs and solid organ transplant recipients (SOTRs). The study was designed in December 2020, and reactogenicity data were obtained during the vaccination campaign conducted at Hospital Clínic of Barcelona (HCB) from 7 January, when a vaccination campaign began in the HCB, until 30 April 2021. The first vaccinee group were HCPs. The type of vaccine was determined by national guidance and availability depending on local supply. Accordingly, at the beginning of the vaccination campaign, BNT162b2 was the only vaccine available for HCPs. The vaccination plan was amended on January 22nd with the inclusion of mRNA-1273 for HCPs, and from March 2nd onwards, mRNA-1273 was also administered to SOTRs. Reactogenicity was collected through a structured questionnaire that was delivered to each vaccinee through a telephone interview after receiving each vaccine dose.

## 2. Materials and Methods

### 2.1. Study Design and Collected Data

This was an observational, unicentric study with real-world data (RWD) that was implemented as a post hoc subanalysis of the VigilVacCOVID study [17]. The population of the VigilVacCOVID study included HCPs vaccinated with BNT162b2 and mRNA-1273 vaccines and SOTRs vaccinated with mRNA-1273. The study was reported following the Strengthening the Reporting of Observational Studies in Epidemiology (STROBE) guidelines for cross-sectional studies [19]. The current post hoc subanalysis was approved by the Ethics Committee of HCB as a major amendment of the VigilVacCOVID study (references: HCB/2021/0684-amendment 1 and HCB/2021/0685-amendment 1). An informed consent waiver was approved by the Ethics Committee given the retrospective nature of the data collection and review, with no interventions performed on patients.

Reactogenicity and tolerability results had been previously obtained during the vaccination campaign in the context of the VigilVacCOVID study. The characteristics of the implemented surveillance program, the methodology of the safety data collection, and the overall reactogenicity profile assessed in the main VigilVacCOVID study are described elsewhere [17].

For the current study, we obtained the immunogenicity results of the above-mentioned participants from the electronic health records of HCB, with the collaboration of the Microbiology and Immunology Departments. The immunogenicity data included humoral (i.e., antibody titres) and cellular immunological results. Humoral immunogenicity had been assessed via the analysis of neutralising and anti-spike antibodies (IgG + IgM), based on the semiquantitative measurement of antibody titres on a scale ranging from 0 to >10. Results of >1.0 were considered positive antibody titres [20]. IgG titres were measured on samples that tested positive in the Cobas platform (Cobas; Roche Molecular Systems, Branchburg, NJ, USA) in a subset of patients, as per local clinical practice protocols [21]. IgG was measured only in a subset of the total vaccinated population (N = 322 vaccinees). Cellular immunogenicity was measured as the development of T cell-virus-specific responses as assessed via the interferon-gamma ELISPOT assay applied to isolated peripheral blood mononuclear cells (PBMCs) [22]. Given that cellular immunogenicity assessment was not routine clinical practice at the time of the vaccination campaign, data on cellular immunity were only available for a subset of vaccinees (N = 59 vaccinees). The complete method description is provided in the Appendix A.

### 2.2. Statistical Analysis

#### 2.2.1. Sample Size Calculation and Planned Analyses

The VigilVacCOVID cohort comprised HCPs vaccinated with BNT162b2 and mRNA-1273 vaccines and SOTRs vaccinated with mRNA-1273. It included a total of 6865 subjects, of whom 5088 responded to the questionnaire and thereby provided reactogenicity data for the current study. Of these, immunogenicity data were available for 3563 participants.

Given that a prospective analysis of immunogenicity had not been planned during our previous VigilVacCOVID study and considering the retrospective and real-world nature of data collection, there were some constraints that we had to take into consideration for the analysis of results. On one hand, the method previously used to measure humoral immunogenicity in clinical practice was semiquantitative and did not distinguish between infection-related (nucleocapsid protein (NCP) IgG antibodies) and vaccine-related (neutralising IgG antibodies to the receptor-binding protein of the S1 subunit of the spike protein) neutralising antibodies. SARS-CoV-2 tests were not routinely performed before and after vaccination. Likewise, in this initial population with available immunogenicity data (N = 3563), some immunity assessments had been performed either within a 4-month period after vaccination or prior to vaccination. We considered that the positive antibody titres of these distant measurements were not representative of ARs occurring immediately after receiving vaccine doses, nor were they necessarily related to vaccination. For instance, the possibility exists that some of the included participants developed immunogenicity before or after vaccination because of SARS-CoV-2 infection and not specifically due to vaccination.

To adjust these potential biases, we decided to perform two sensitivity analyses (see flowchart in Figure 1). We first refined the initial population by restricting the immunogenicity assessment timeframe to 4 weeks, and we defined subpopulation 2 for sensitivity analysis 2 (N = 597). Furthermore, we excluded vaccinees who had immunogenicity data prior to vaccination (N = 215). This was considered the most sensitive and least-biased population, and for this reason, it was considered the main population for analysis. We subsequently redefined the initial population (N = 3563) as subpopulation 1, and we considered the analysis of this population as sensitivity analysis 1. The main aim of this sensitivity analysis was to assess the consistency of results across analyses with a higher sample size. Baseline characteristics of subpopulations 1 and 2 are displayed in the Appendix A, and the baseline characteristics of the main population are depicted in Table 1.

#### 2.2.2. Study Endpoints

The primary endpoint was to assess the association between reactogenicity after vaccination and the development of humoral immunogenicity in the main population. Reactogenicity was defined in terms of the proportion of vaccinees with mild, moderate, and severe ARs for each immunogenicity category and in terms of AR intensity.

The key secondary endpoint was to assess the association between specific solicited and unsolicited AR and the development of immunogenicity.

As exploratory endpoints, we assessed the same reactogenicity and humoral immunogenicity endpoints as in sensitivity analyses 1 and 2. In addition, cellular immunogenicity was assessed as an exploratory endpoint in subpopulation 1. Cellular immunogenicity was not assessed in the main population, nor in subpopulation 2, given the lack of sample size and available data on cellular immunogenicity.

Finally, reactogenicity and humoral immunogenicity were assessed, in a descriptive and exploratory manner, in different subgroups of subpopulation 1, to assess potential trends that could suggest a difference with regard to the type of administered vaccine (mRNA-1273 vs. BTN162b2) or the vaccinated population (HCP vs. SOTR). Due to the sample size limitations and its exploratory nature, this descriptive analysis was only performed on subpopulation 1 and only considered the humoral immunogenicity results.

#### 2.2.3. Statistical Analysis

Categorical variables were analysed as frequencies and percentages, whereas continuous variables were described as the mean ± standard deviation or median (25–75% interquartile range), as deemed suitable. Categorical data underwent comparison via the chi-square test or the Fisher’s exact test, as applicable. Continuous variables were assessed using Mann–Whitney testing.

Reactogenicity data were expressed in terms of the solicited and unsolicited AR rate (n, %), types of solicited and unsolicited ARs, intensity distribution of solicited ARs according to three grades of maximum intensity (mild, moderate, severe), and the mean and median Likert score (LS) values of the solicited ARs. Intensity was assessed as the proportion of vaccinees with mild, moderate, and severe ARs for each immunogenicity category (positive or negative). In addition, the mean (SD) Likert score (LS) (with values ranging from 0 to 10) of the intensity of the developed ARs was assessed for each immunogenicity category. Finally, the distribution of AR intensity in each immunogenicity category was analysed with the maximum intensity grade of AR experienced after vaccination and the distribution (%) of vaccinees in each maximum grade according to whether they tested positive or negative in the immunogenicity tests. These analyses were performed after the first, second, or any vaccine dose. Severe ARs were defined by an LS score of 7–10. Intensity was measured for solicited ARs, which were the primary events for study. Due to the subjectivity of some solicited ARs and the questionnaire-based methodology of the study, intensity was not measured for insomnia, nausea, diarrhoea, and vomiting, or for unsolicited ARs that had been collected through a free text option.

Given that specific IgG titres were only measured in 322 vaccinees, and on a different platform than the one used for total antibody titres, we decided to use only the total semiquantitative antibody titres for the immunogenicity analysis and not IgG titres. For the analysis of immunogenicity, we defined antibody positiveness considering a cut-off of >1.0 (positive humoral immunogenicity above a cut-off of >1.0; for the cut-off and detection method, see the Materials and Methods and Appendix A). Given that antibody titres were measured with a semiquantitative method on a scale ranging from 0 to >10, we calculated the median value of the titres’ level (second quartile, Q2), and then analysed the distribution of vaccinees above and below the median to carry out a semiquantitative assessment that considered the extent of elicited humoral immunogenicity. Results were expressed as absolute and relative numbers of vaccinees with the reported AR and negative antibody titres or titres below the median value vs. vaccinees with the reported AR and positive antibody titres or titres above the median value, with *p*-values. Cellular immunogenicity was not assessed in the main population, nor in subpopulation 2, given the lack of sample size and available data on cellular immunogenicity (N = 59) and because this was considered as an exploratory analysis.

We applied a two-sided significance level of 5%. All calculations were performed using SAS v9.4 software (Cary, NC, USA). In addition, we conducted exploratory analyses using a less conservative significance level of 10% to explore, in a descriptive manner, any potential relationships that could be of clinical interest despite not meeting the standard 5% threshold.

## 3. Results

### 3.1. Immunogenicity and Reactogenicity Results

The overall results are depicted in Table 2, Table 3 and Table 4. The individual absolute and relative numbers of each assessed variable can be found in the tables in the Appendix A. Figure 2, Figure 3 and Figure 4 depict the association between immunogenicity and reactogenicity (as a percentage of solicited ARs). For the purposes of clarity and to highlight the main results of the analyses, only those ARs that showed a significant association are displayed. The complete graphic results of Table 2, Table 3 and Table 4 are shown in the Appendix A.

#### 3.1.1. Main Analysis (Main Population, N = 215)

##### Immunogenicity Assessed as Positive/Negative Antibody Titres

We did not observe any significant association with the maximum intensity grades or LS (mean, SD). Regarding specific ARs, we observed a significant association between the development of severe malaise after any dose and the development of positive antibody titres against SARS-CoV-2 (vaccinees with malaise: 1 (1.6%) with negative antibody titres, 15 (9.9%) with positive antibody titres, *p* = 0.033) (Figure 2, Table 2, and Appendix A).

##### Immunogenicity Assessed as Antibody Titres above/below Median

No significant associations were observed in the intensity analysis (Table 2 and Appendix A).

Regarding specific ARs, we observed a significant association between the development of fever (after first dose: 5 (5.4%) vs. 19 (15.6%), *p* = 0.019; after any dose: 0 (0.0%) vs. 7 (5.7%), *p* = 0.019; after second dose: 0 (0.0%) vs. 5 (4.1%), *p* = 0.048), arm pain (after first dose: 10 (10.8%) vs. 26 (21.3%), *p* = 0.019), lymphadenopathy (after any dose: 0 (0.0%) vs. 6 (4.9%), *p* = 0.030; after second dose: 0 (0.0%) vs. 5 (4.1%), *p* = 0.048), and the measurement of semiquantitative antibody titres above the median. These ARs developed after any vaccine dose and did not demonstrate a specific predominance in frequency over the first or second doses (Figure 2, Table 2, and Appendix A).

In terms of severe ARs, we observed a significant association with the development of severe fever (after any dose: 0 (0.0%) vs. 7 (5.7%), *p* = 0.019; after second dose: 0 (0.0%) vs. 5 (4.1%), *p* = 0.048) (Table 2 and Appendix A).

#### 3.1.2. Sensitivity Analysis 1 (N = 3563)

##### Immunogenicity Assessed as Positive/Negative Antibody Titres

A significant association was found in the distribution of ARs of grades 1–3 after any vaccine dose (grade 1: 15 (23.8% vs. 350 (12.2%); grade 2: 23 (36.5%) vs. 1132 (39.3%); grade 3: 25 (39.7%) vs. 1397 (48.5%), *p* = 0.020) and after a second vaccine dose (grade 1: 12 (27.3%) vs. 261 (11.6%); grade 2: 19 (43.2%) vs. 949 (42.3%); grade 3: 13 (29.5) vs. 1036 (46.1%, *p* = 0.003)). We also found a significant association with intensity measured by LS (mean, SD) after any vaccine dose (5.48 (2.47%) vs. 6.15 (2.36%), *p* = 0.031) and after a second vaccine dose (4.95 (2.37%) vs. 6.10 (2.06%), *p* = 0.001). Likewise, there was a significantly higher proportion of severe ARs after a second dose (13 (17.1%) vs. 1036 (29.7%), *p* = 0.017) (Table 3, and Appendix A).

Regarding specific ARs, we observed a significantly higher proportion of fever after a second dose (11 (14.5%) vs. 974 (27.9%), *p* = 0.010), malaise after any dose (7 (9.2%) vs. 622 (17.8%), *p* = 0.050) and after a second dose (4 (5.3%) vs. 517 (14.8%), *p* = 0.020), chills after a second dose (2 (2.6%) vs. 385 (11.0%), *p* = 0.020), myalgia after any dose (5 (6.6%) vs. 709 (20.3%), *p* = 0.003) and after a second dose (4 (5.3%) vs. 621 (17.8%), *p* = 0.004), and headache after any dose (11 (14.5%) vs. 967 (27.7%), *p* = 0.010) and after a second dose (8 (10.5%) vs. 750 (21.5%), *p* = 0.020), (Appendix A). Regarding severe specific ARs, a higher proportion of these were observed for severe malaise after any dose (1 (1.2%) vs. 305 (8.8%), *p* = 0.010), severe malaise after a second dose (1 (1.2%) vs. 258 (7.4%), *p* = 0.030), severe headache after any dose (3 (3.9%) vs. 419 (12.0%), *p* = 0.030), and severe headache after a second dose (2 (2.6%) vs. 331 (9.5%), *p* = 0.040) (Figure 3, Table 3, and Appendix A).

##### Immunogenicity Assessed as Antibody Titres above/below Median

We observed a significant association for ARs of grades 1–3, LS (mean, SD), and the development of any AR after a second vaccine dose (Appendix A).

As for specific ARs, we observed a significantly higher proportion of chills after a second dose (35 (8.0%) vs. 352 (11.3%), *p* = 0.040) as well as severe headache after any dose (39 (8.9%) vs. 383 (12.3%), *p* = 0.040) and severe headache after a second dose (28 (6.4%) vs. 305 (9.8%), *p* = 0.020) in vaccinees with antibody titres above the median cut-off (Table 3 and Appendix A).

##### Cellular Immunogenicity

A positive trend of a higher frequency of vaccinees with positive cellular immunogenicity was observed for some solicited ARs such as malaise, chills, arm pain, myalgia, and severe headache (Figure 3 and Appendix A). However, none of these associations were statistically significant. The analysis of intensity did not show any differences between vaccines with negative and positive cellular immunogenicity.

##### Descriptive Analysis According to Type of Vaccine and Population

Regarding vaccine type (mRNA-1273 vs. BNT162b2), data obtained from HCPs vaccinated with both types of vaccine showed similar immunogenicity rates (98.5% vs. 99% of vaccinees with positive antibody titres, respectively). However, the reactogenicity seemed to be greater with mRNA-1273 vaccines than with BNT162b2 (the proportion of any AR after vaccination was 97.4% vs. 81.7%, respectively), especially concerning severe AR rates (61.8% vs. 34.2%, respectively). Increased reactogenicity with mRNA-1273 vaccines was also observed in the analysis of solicited ARs (Appendix A).

We also observed a difference in terms of immunogenicity and reactogenicity between the HCPs and SOTRs after vaccination with mRNA-1273 vaccines. Accordingly, we observed a decreased rate of immunogenicity in SOTRs as opposed to HCPs (57% vs. 98.5% of vaccinees with positive antibody titres, respectively) and a decreased reactogenicity rate (proportion of vaccinees with at least one AR, 87.4% vs. 97.4%, respectively; proportion of vaccinees with at least one severe AR, 28.2% vs. 61.8%, respectively). The same trend was observed with the solicited ARs (Appendix A).

#### 3.1.3. Sensitivity Analysis 2 (N = 597)

##### Immunogenicity Assessed as Positive/Negative Antibody Titres

Although there was a positive trend, we did not find a significant association with ARs of grades 1–3 after the first vaccine dose. There was a significant association with intensity measured as the LS after a second dose (3.63 (3.31%) vs. 3.01 (3.23%), *p* = 0.001). In addition, we observed a positive association between the development of any AR after a second dose (187 (68.5%) vs. 188 (58.0%), *p* = 0.008) and the development of any severe AR after the first dose (53 (19.4%) vs. 90 (27.8%), *p* = 0.017) (Table 4 and Appendix A).

As for the distribution of specific ARs, we observed a significant association with fever after a first dose (16 (5.9%) vs. 73 (22.5%), *p* ≤ 0.001), malaise after any dose (37 (13.6%) vs. 76 (23.5%), *p* ≤ 0.002) and after a first dose (5 (1.8%) vs. 36 (11.1%), *p* ≤ 0.001), chills after a first dose (2 (0.7%) vs. 25 (7.7%), *p* ≤ 0.001), myalgia after a first dose (9 (3.3%) vs. 36 (11.1%), *p* = 0.003), arthralgia after a first dose (0 (0.0%) vs. 10 (3.1%), *p* = 0.034), and headache after a first dose (24 (8.8%) vs. 48 (14.8%), *p* = 0.024) (Table 2 and Appendix A). Regarding severe ARs, severe fever after a first dose (0 (0.0%) vs. 11 (3.4%), *p* = 0.002), severe malaise after any dose (17 (6.2%) vs. 42 (13.0%), *p* = 0.006) and after a first dose (3 (1.1%) vs. 18 (5.6%), *p* = 0.003), severe myalgia after a first dose (3 (1.0%) vs. 18 (5.8%), *p* = 0.003), severe arthralgia after a first dose (0 (0.0%) vs. 6 (1.9%), *p* = 0.024), and severe headache after a first dose (6 (2.2%) vs. 20 (6.2%), *p* = 0.018) showed a significant association with the development of positive antibody titres (Figure 4, Table 4, and Appendix A).

##### Immunogenicity Assessed as Antibody Titres above/below Median

We did not observe any significant association with the maximum intensity grades or with LS (mean, SD). We observed a significant association with the proportion of ARs after a second dose (193 (67.0%) vs. 182 (58.9%), *p* = 0.040) and the proportion of any moderate AR after the first dose (103 (35.8%) vs. 135 (43.7%), *p* = 0.048) (Table 4 and Appendix A).

A positive association was also observed for some specific ARs such as fatigue after the first dose (24 (8.3%) vs. 43 (13.9%), *p* = 0.031), fever after the first dose (19 (6.6%) vs. 70 (22.7%), *p* ≤ 0.001), malaise after any dose (41 (14.2%) vs. 72 (23.3%), *p* = 0.004) and after the first dose (7 (2.4%) vs. 34 (11.0%), *p* ≤ 0.001), chills after a first dose (4 (1.4%) vs. 23 (7.4%), *p* = 0.004), myalgia after a first dose (9 (3.1%) vs. 36 (11.7%), *p* ≤ 0.001), arthralgia after the first dose (1 (0.3%) vs. 9 (2.9%), *p* = 0.015), and headache after a first dose (26 (9.0%) vs. 46 (14.9%), *p* = 0.028). Regarding severe ARs, we observed a significant association for severe fever after the first dose (1 (0.3%) vs. 10 (3.2%), *p* = 0.009), severe malaise after any dose (19 (6.6%) vs. 40 (12.9%), *p* = 0.009) and after the first dose (5 (1.7%) vs. 16 (5.2%), *p* = 0.023), severe myalgia after the first dose (3 (1.0%) vs. 18 (5.8%), *p* = 0.002), severe arthralgia after the first dose (1 (0.3%) vs. 5 (1.6%), *p* = 0.120), and severe headache after a first dose (6 (2.1%) vs. 20 (6.5%), *p* = 0.009) (Table 4 and Appendix A).

#### 3.1.4. Overall Reactogenicity and Immunogenicity Analysis

Table 2, Table 3 and Table 4 depict ARs for which statistical significance or a trend towards significance (with a more conservative alpha level of 10%) was observed. After a consistency assessment among the different analyses, a higher proportion of fever, malaise, and myalgia including severe episodes of these ARs was observed among vaccinees that developed positive antibody titres in the three analyses. The proportion of vaccinees with myalgia was statistically significant in sensitivity analyses 1 and 2, and a trend towards significance (alpha level of <10%) was observed for severe myalgia when antibody positiveness was considered in the main analysis. These ARs developed after any vaccine dose in the primary analysis and sensitivity analysis 1 and did not demonstrate a specific predominance in frequency over the first or second doses, whereas most of the statistically significant ARs in sensitivity analysis 2 were observed after any dose or after a second vaccine dose. Figure 2, Figure 3 and Figure 4 show an association between reactogenicity (as expressed by solicited ARs) and immunogenicity that seemed consistent across the three performed analyses.

Some ARs such as chills and headache were significant in sensitivity analyses 1 and 2 but were not significant in the main analysis. Similarly, the association between the intensity of ARs and immunogenicity was not consistent either.

Of note, in some of these ARs (e.g., injection site pain, injection site swelling, fatigue, hypersensitivity, gastrointestinal disorders), a significant association was observed, but this association was observed in a higher proportion of vaccinees from the negative immunogenicity group, thereby showing a negative association. The individual absolute and relative numbers of these ARs can be found in the tables in the Appendix A.

## 4. Discussion

We believe our study provides novel insights into the enquiry of the hypothesis of the potential link between immunogenicity and reactogenicity in mRNA vaccines including the following: (i) an exploration of both humoral and cellular immunogenicity, (ii) a methodology for assessing both the development and quantity of immunogenicity, (iii) the inclusion of a large vaccinated cohort composed of HCPs and immunocompromised patients that can be more reflective of typical vaccination practices, (iv) the assessment of two mRNA vaccines, and (v) the evaluation of a wide range of local and systemic reactions with the additional consideration of symptom intensity.

Overall, we observed a consistent significant association between fever, malaise, and myalgia and the development of elicited humoral immunogenicity. According to our data, these results also seem to be associated with the quantity of immunogenicity developed. In the sensitivity analyses performed, we also observed a significant association between chills, headache, injection site pain, arm pain, arthralgia, and elicited immunogenicity.

On the whole, our results are consistent with those reported in other studies by various authors. Interestingly, most of these previous works studied HCPs vaccinated with BNT162b2 vaccines. Speletas et al. studied the intensity and duration of immunogenicity in 511 individuals who were vaccinated with two doses of the BNT162b2 vaccine. Taking into account the impact of adverse side effects on the strength of IgG antibody responses, a multivariate analysis showed that on day 21, only increased age had a significant effect, with no impact observed from AR. However, individuals who experienced fever and muscle pain after the second dose had notably higher IgG levels on day 42. Interestingly, this effect had disappeared by day 90 and day 180 following vaccination [23].

Yamamoto et al. assessed the immunogenicity and reactogenicity in 100 HCPs who had been vaccinated with BNT162b2. Participants who exhibited fever showed notably elevated spike IgG titres 7 days after the second vaccination dose compared to those who reported no fever or only mild fever. Although spike IgG titres tended to decline over time for all participants, those who had a high fever still maintained higher titres than those who experienced no or mild fever, even at both 39 and 60–74 days post-vaccination [24]. Other studies have also found a positive association between local and systemic ARs and the development of immunogenicity after vaccination with BNT162b2 [25,26].

Immunogenicity and reactogenicity after vaccination with mRNA-1273 vaccines have been explored less than with BNT162b2. Matsumoto et al. assessed the dynamics of immunogenicity in 47 healthy individuals vaccinated with a third dose of mRNA-1273 at 3 days, 7 days, and 1 month post-vaccination. The investigators found a significant association between fever and the development of antibody titres [27]. Other SARS-CoV-2 vaccines have also been studied in terms of immunogenicity and reactogenicity, without consistent results. Yun Lim et al. assessed a prospective cohort of HCPs vaccinated with ChAdOx1 and BNT162b2. In their study, only a positive association with local ARs was observed [28].

Alternatively, other authors have found negative results after vaccination with ChAdOx1 and BNT162b2, with sample sizes of 67 HCP [29], 80 HCP [15], and 135 healthy adults [30], while others have only observed some association with AR intensity but not with AR frequency [14], a weak correlation [31], or an association only in males but not in female vaccinees [32].

With regard to AR intensity, while we found some significant differences in the sensitivity analyses, we did not observe any difference in the main population of the study, as was the case in most of the previously described studies. Hence, a positive association with AR intensity cannot be firmly inferred.

Compared to previously reported results, our study analysed a wider study population (HCP and SOTR), two mRNA vaccines (BNT162b2 vs. mRNA-1273), and both immunity arms (humoral and cellular)). In the explanatory analysis, we observed decreased immunogenicity and reactogenicity in SOTRs vaccinated with mRNA-1273 in contrast to HCPs who had also been vaccinated with mRNA-1273. While reactogenicity was higher with mRNA-1273 in the vaccinated HCPs, and taking into consideration that certain reactogenicities may stem from vaccine composition rather than immunogenicity itself, the differences observed in reactogenicity between HCPs and SOTRs provide evidence for the role of immunogenicity. These data, together with the results in the overall population, provide support for the confirmation of our initial hypothesis formulated following the results obtained in our first study.

Notwithstanding, our study has several limitations. Firstly, our study was based on reactogenicity data from the VigilVacCOVID study. Therefore, the AR data may have limitations inherent to the observational and non-randomised nature of the VigilVacCOVID study such as recall bias and potential confounders. In addition, AR data were obtained through a structured questionnaire. This can be a subjective method for the assessment of certain ARs such as fatigue or malaise. Furthermore, the method employed for assessing humoral immunogenicity in clinical practice was semiquantitative and did not differentiate between infection-related and vaccine-induced neutralising antibodies. Routine SARS-CoV-2 testing was not conducted pre- and post-vaccination. Additionally, some immunogenicity measurements were obtained months after vaccination, while others were obtained prior to vaccination, potentially not reflecting ARs immediately following vaccination. Therefore, we cannot exclude the possibility that some participants may have developed immunogenicity due to SARS-CoV-2 infection rather than vaccination. However, we performed two sensitivity analyses to account for these potential biases; this reinforced the consistency of our main results.

Of note, the first sensitivity analysis included all vaccinees followed-up in the VigilVacCOVID study with immunogenicity and reactogenicity data (3563 vaccines); these data were collected in a 4-month timeframe. Despite the limitations related to the 4-month study timeframe, one should take into consideration that previous authors have found an association between ARs and immunogenicity even at 74 days post-vaccination [24]. Therefore, we consider that the results obtained in the 4-month timeframe, with a higher sample size of 3563 vaccinees, are still informative. The results of cellular immunogenicity were negative, both in the rate and intensity of ARs and in the assessment of solicited and unsolicited AR. We acknowledge that our sample size for this assessment was rather small (n = 59 vaccinees with testing performed), given that cellular immunogenicity was not standard practice for all vaccinees at the time of the vaccination campaign. Most likely because of the sample size, we could observe some positive trends in some solicited ARs, but none were statistically significant. Although Speletas et al. elucidated a clear correlation between humoral and cellular responses [23], few studies have assessed both humoral and cellular immunogenicity, and those that did measure the T cell responses included similar sample sizes [15,28,32] with negative results. Our results, in terms of cellular immunogenicity, were statistically non-significant. However, we observed a positive trend, with 86% of vaccinees testing positive for cellular immunogenicity amongst those with positive antibody titres. A relevant aspect to consider is that humoral immunogenicity is a surrogate marker of the T cell response; in fact, there is no immune memory without T lymphocytes and, although it is not frequent, there can be one without antibodies or B lymphocytes [33]. Although the humoral response can be considered a valid marker of the association between reactogenicity and immunogenicity, it is relevant to assess the two arms of adaptive immunity (humoral and cellular immunogenicity) whenever possible, especially in those patients without an antibody response and/or at risk of severe COVID-19. Lack of standardised protocols for the concomitant assessment of the two arms of adaptive immunity in the standard practice of clinical centres can be a limitation for the collection of RWD regarding cellular immunogenicity, and humoral immunogenicity can be a valid marker in these scenarios. Nonetheless, further studies with larger sample sizes and where reactogenicity and cellular immunogenicity are assessed are necessary, especially with the likely development of new vaccines based on the mRNA platform.

On the other hand, a significant association was observed for some ARs, but this association was found among a higher proportion of vaccinees in the negative immunogenicity group. The clinical meaning of this finding is unclear to us, and it should be explored in future studies where systemic and local ARs are thoroughly assessed.

Moreover, immunogenicity assessments were based solely on the total antibody titres (IgM + IgG), and specific IgG titres were only measured in a subset of vaccinees. While we are aware of the limitations related to data sources, data type, and the timing of determinations, we would like to highlight that these limitations are due to RWD acquisition processes. Our data are reflective of the standard practice during the period when the vaccination campaign was established rapidly amid the pressures of SARS-CoV-2 fears, and safety follow-ups were also organised rapidly to provide vaccinees with exhaustive vaccine surveillance. Immunogenicity assessments were not standard practice at the time and, as such, could not be carried out during the vaccinations. Because the data were obtained within the limitations of standard practice, we consider the results to be representative, and that these as results can be generalised to the overall population who are routinely vaccinated at healthcare centres. The methodology of our study reflects data acquisition and processing within a real-world context in standard clinical practice, which should be instructive for the development of further RWD studies, especially those that are performed within the context of mass vaccination campaigns during emergency periods with the use of data from clinical practices.

Translating our results into clinical practice, fever, myalgia, and malaise are systemic symptoms that develop rapidly after vaccination and can be relatively easily detected. Furthermore, these symptoms are relatively frequent (the European Medicines Agency has identified 1.7 million spontaneous reports of suspected ARs in almost 768 million vaccine doses administered in Europe [34], of which fever and myalgia are two of the most reported ARs in the European database). The reporting of these symptoms may be indicative of the prospective development of immunogenicity after vaccination. Likewise, SOTRs may be asymptomatic after vaccination and may have lower chances of developing immunogenicity. The main focus of our study was to assess a potential link between reactogenicity and immunogenicity. These are preliminary results that cannot be translated directly into clinical practice. That is to say, immunogenicity is a heterogenous multimodal clinical entity that can be explained by several factors other than reactogenicity such as genomic data or other types of clinical data, among others. The construction and validation of a future prediction signature may be helpful when planning immunogenicity testing programs, vaccination campaigns, and closer monitoring programs in special patient subgroups in clinical care centres. Therefore, we consider that the data of our study, which revealed an association between immunogenicity and reactogenicity, can contribute to the creation of this signature in further dedicated studies.

Conversely, the EMA Pharmacovigilance Risk Assessment Committee (PRAC) began an analysis of rare cases of myocarditis and pericarditis after vaccination with BNT162b2 [35] following early reports of cases of myocarditis after vaccination with BNT162b2 [36]. The safety signal of myocarditis and pericarditis was finally validated and included in the Summary of Product Characteristics (SmPC) of the vaccines [37]. The overall incidence of myocarditis and pericarditis is very low. Although figures varied according to the different series [38,39,40], published evidence shows that these events tend to be rare or very rare. Therefore, with 5088 subjects, the VigilVacCOVID study was not powered enough to detect rare cases such as myocarditis or pericarditis, but adequate for obtaining a short-term reactogenicity profile. The current study, in line with the VigilVacCOVID study, aimed to characterise the association between short-term reactogenicity and developed immunogenicity.

In addition, apart from myocarditis and pericarditis, no other safety signals related to cardiovascular complications have been validated to date. Cardiovascular events such as myocardial infarctions, strokes, embolism, and others have been studied, and while some studies have suggested increased reporting rates of these events [41,42], others have not found such an association [36,43,44,45,46,47]. While some authors have previously suggested that mRNA vaccines have been associated with increased reports of adverse cardiovascular reactions than other types of SARS-CoV-2 vaccines, it is true that the strongest association remains thus far with myocarditis and pericarditis, and the evidence regarding the association of SARs-CoV-2 vaccines with cardiovascular outcomes still remains to be elucidated [48]. In our previous VigilVacCOVID study, we detected some cases of syncope, hypertension, hypotension, chest pain, cold sweat, and tachycardia as unsolicited ARs; similar findings were previously reported as potential cardiovascular adverse effects in other studies [41]. These results confirm the sensitivity of our study to detect and report ARs developed after vaccination, in line with what has been previously published. However, since the study was designed to obtain a short-term reactogenicity profile, we were not able to confirm whether these symptoms were finally confirmed through a cardiovascular diagnosis, and whether there was a causal link with the vaccine. Therefore, in the current analysis, we did not observe an association between these events and immunogenicity.

Finally, our study suggests a positive association between reactogenicity and the development and quantity of immunogenicity after vaccination. This association seems to be linked to the development of anti-spike protective immunogenicity and should not be generalised to other types of adaptive immunity. For instance, recent reports have suggested that autoantibodies targeting type I IFNs develop after vaccination with mRNA SARS-CoV-2 vaccines but not with viral vector-based vaccines [49,50]. While these autoantibodies have been associated with adverse reactions in other vaccines such as the live-attenuated yellow vaccine [51], such an association has not been observed in SARS-CoV-2 vaccines [52,53].

## 5. Conclusions

In conclusion, our study detected a positive association between reactogenicity (defined as the onset of systemic symptoms such as fever, myalgia, and malaise) and the development and quantity of immunogenicity after vaccination. In addition to their high prevalence, these ARs are early, easily detectable symptoms post-vaccination and are indicative of the development of potential immunogenicity. These results can be valuable for the future construction of immunogenicity prediction signatures. Moreover, we believe that our results can be impactful in advancing our understanding of the immunological correlates of mRNA vaccines and, consequently, in the design of future mRNA vaccine constructs.

## Figures and Tables

**Figure 1 vaccines-12-00665-f001:**
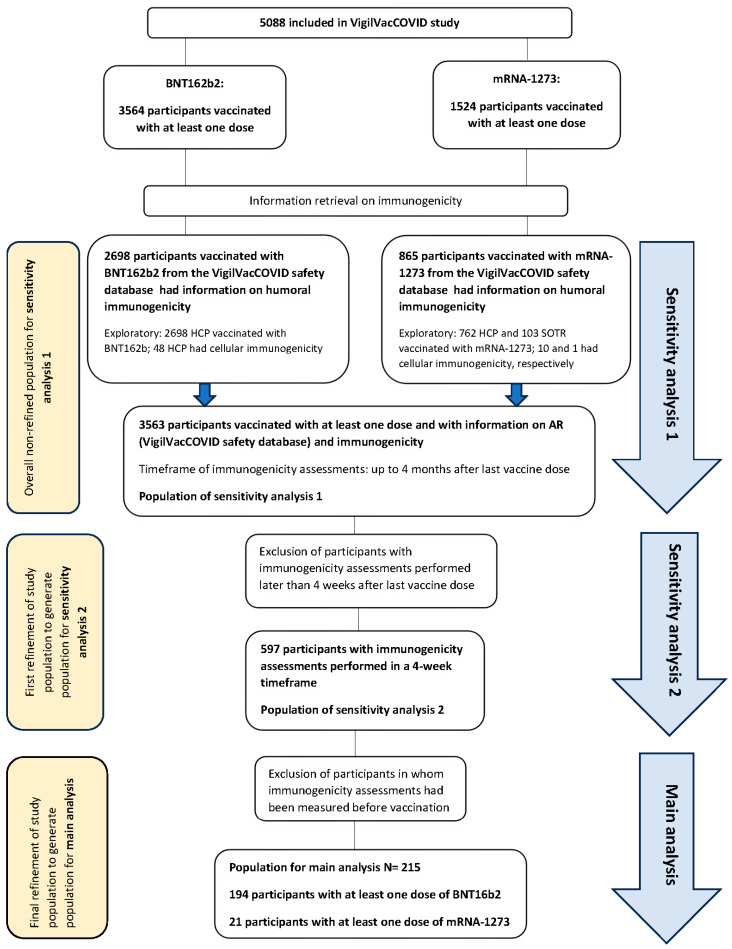
Flowchart of the study displaying the populations for the main analysis and sensitivity analyses.

**Figure 2 vaccines-12-00665-f002:**
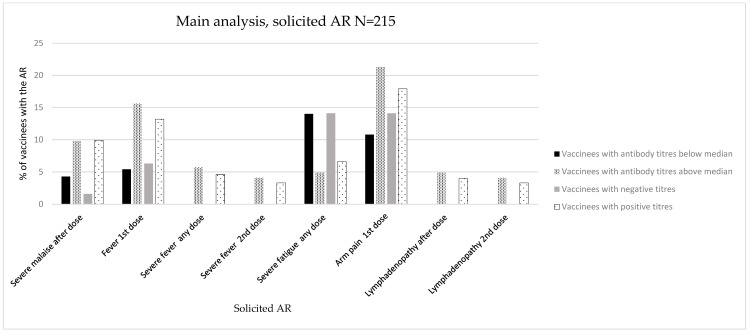
Immunogenicity and reactogenicity results of the main population. Bar diagram of the results in Table 2. Immunogenicity is expressed as antibody positiveness (positive/negative titres) and antibody titres (antibody titres below/above median) data (%). Reactogenicity results are represented as a percentage of vaccinees developing each solicited AR.

**Figure 3 vaccines-12-00665-f003:**
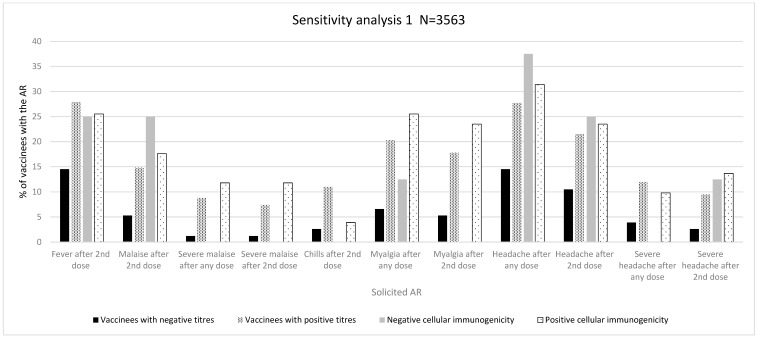
Immunogenicity and reactogenicity results of sensitivity analysis 1. Bar diagram of the results in Table 3. Immunogenicity is expressed as antibody positiveness (positive/negative titres) and cellular immunogenicity (positive/negative) data (%). Reactogenicity results are represented as a percentage of vaccinees developing each solicited AR.

**Figure 4 vaccines-12-00665-f004:**
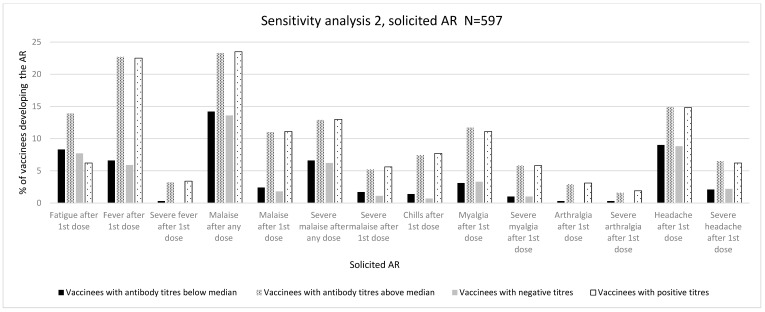
Immunogenicity and reactogenicity results of sensitivity analysis 2. Bar diagram of the results in Table 4. Immunogenicity is expressed as antibody positiveness (positive/negative titres) and antibody titres (antibody titres below/above median) data (%). Reactogenicity results are represented as a percentage of vaccinees developing each solicited AR.

**Table 1 vaccines-12-00665-t001:** Characteristics of the participants (main analysis).

Covariates	Total Population (N = 215)
**Age**	
N	215
Mean (SD)	41.13 (12.71)
**Gender**	
Male (n, %)	70 (32.6)
Female (n, %)	145 (67.4)
**Occupational SARS-CoV-2 contact**	
In contact with SARS-CoV-2 patients	105 (48.8)
Without contact with SARS-CoV-2 patients	110 (51.2)
**^a^** **Previous SARS-CoV-2 infection**	
Yes	78 (36.3)
No	137 (63.7)
**Comorbidities**	
^b^ Any comorbidity	83 (38.6)
Arterial hypertension	12 (5.6)
Diabetes mellitus	3 (1.4)
Heart failure	1 (0.5)
Asthma	8 (3.7)
Rheumatic/immune-mediated disease	1 (0.5)
**Drug allergies**	17 (7.9)
**Food allergies**	10 (4.7)
**Type of administered vaccine**	
BNT16b2	
First dose:	194 (90.2)
Second dose:	138 (64.2)
mRNA-1273	
First dose:	21 (9.8)
Second dose:	17 (7.9)
**Humoral immunogenicity (antibody positiveness on semiquantitative antibody titres)**	
Number of vaccinees with positive antibody titres (titres of >1.0) (n, %)	151 (70)
Number of vaccinees with negative antibody titres (titres of <1.0) (n, %)	64 (30)
**Humoral immunogenicity (antibody semiquantitative titres above/below median)**	
Median value of antibody titres (P25, P75) *	7.69 (0.73,10.00)
Number of vaccinees with antibody titres above median (n, %)	93 (43%)
Number of vaccinees with antibody titres below median (n, %)	122 (57%)

Covariables of studied population. ^a^ Diagnosis via RCP, antigen test, and antibody positive serology. ^b^ Number of vaccinees with at least one comorbidity; * On a semiquantitative scale ranging from 0 to >10.

**Table 2 vaccines-12-00665-t002:** Results of the main analysis.

Main Analysis							
N = 215		Antibody Positiveness		*p*-Value	Antibody Titres		*p*-Value
		Negative Antibody Titres (n = 64)	Positive Antibody Titres (n = 151)		Antibody Titres below Median * (n = 93)	Antibody Titres above Median * (n = 122)	
Proportion of AR (% of vaccinees with an AR)	Any mild AR after 1st dose (%)	19 (29.7)	28 (18.5)	0.071 ^n^	22 (23.7)	25 (20.5)	0.578
Proportion of specific AR (% of vaccinees with an AR)	Malaise after any dose (%)	4 (6.3)	24 (15.9)	0.055 ^n^	9 (9.7)	19 (15.6)	0.203
	Malaise after 2nd dose (%)	3 (4.7)	19 (12.6)	0.081 ^n^	8 (8.6)	14 (11.5)	0.491
	**Severe malaise after any dose (%)**	**1 (1.6)**	**15 (9.9)**	**0.033**	4 (4.3)	12 (9.8)	0.126
	Severe malaise after 2nd dose (%)	1 (1.6)	13 (8.6)	0.056 ^n^	4 (4.3)	10 (8.2)	0.251
	**Fever after 1st dose (%)**	4 (6.3)	20 (13.2)	0.136	**5 (5.4)**	**19 (15.6)**	**0.019**
	**Severe fever after any dose (%)**	0 (0.0)	7 (4.6)	0.080 ^n^	**0 (0.0)**	**7 (5.7)**	**0.019**
	**Severe fever after 2nd dose (%)**	0 (0.0)	5 (3.3)	0.141	**0 (0.0)**	**5 (4.1)**	**0.048**
	Severe fatigue after any dose (%)	9 (14.1)	10 (6.6)	0.079 ^n^	13 (14.0)	6 (4.9)	0.020
	Arm pain after any dose (%)	15 (23.4)	38 (25.2)	0.788	17 (18.3)	36 (29.5)	0.058 ^n^
	Arm pain after 1st dose (%)	9 (14.1)	27 (17.9)	0.493	10 (10.8)	26 (21.3)	0.040
	Lymphadenopathy after any dose (%)	0 (0.0)	6 (4.0)	0.106	0 (0.0)	6 (4.9)	0.030
	Lymphadenopathy after 2nd dose (%)	0 (0.0)	5 (3.3)	0.141	0 (0.0)	5 (4.1)	0.048
	Severe myalgia after 1st dose (%)	2 (3.1)	7 (4.6)	0.613	0 (0.0)	4 (3.3)	0.078 ^n^
	Severe injection site swelling after any dose (%)	0 (0.0)	4 (2.6)	0.189	0 (0.0)	4 (3.3)	0.078 ^n^
	Severe injection site swelling after 1st dose (%)	0 (0.0)	4 (2.6)	0.189	0 (0.0)	4 (3.3)	0.078 ^n^

NS: Non-significant; *p* ≤ 0.05: statistically significant; statistically significant results that are consistent in the three analyses (overall immunogenicity analysis) are shown in bold type; ^n^ ARs for which a positive nominal trend was observed; * Median value of antibody titres (P25, P75) = 7.69 (0.73, 10.00) on a semiquantitative scale ranging from 0 to >10.

**Table 3 vaccines-12-00665-t003:** Results of sensitivity analysis 1.

Sensitivity Analysis 1							
N = 3563		Antibody Positiveness		*p*-Value	Antibody Titres		*p*-Value
		Negative Antibody Titres (n = 76)	Positive Antibody Titres (n = 3487)		Antibody Titres below Median * (n = 440)	Antibody Titres above Median * (n = 3123)	
Intensity (maximum intensity grade; % vaccinees at each grade)	Grade 1 after any vaccine dose (%)	15 (23.8)	350 (12.2)	0.020	62 (16.9)	303 (11.8)	0.017
	Grade 2 after any vaccine dose (%)	23 (36.5)	1132 (39.3)		142 (38.7)	1013 (39.3)	
	Grade 3 after any vaccine dose (%)	25 (39.7)	1397 (48.5)		163 (44.4)	1259 (48.9)	
	Grade 1 after 2nd vaccine dose (%)	12 (27.3)	261 (11.6)	0.003	47 (17.0)	226 (11.2)	0.006
	Grade 2 after 2nd vaccine dose (%)	19 (43.2)	949 (42.3)		121 (43.8)	847 (42.1)	
	Grade 3 after 2nd vaccine dose (%)	13 (29.5)	1036 (46.1)		108 (39.1)	941 (46.7)	
Intensity (mean, SD Likert score)	After any dose (mean, SD)	5.48 (2.47)	6.15 (2.36)	0.031	5.91 (2.17)	6.16 (2.38)	0.067 ^n^
	After 2nd dose (mean, SD)	4.95 (2.37)	6.10 (2.06)	0.001	5.64 (2.14)	6.14 (2.05)	0.003
Proportion of ARs (% of vaccinees with an AR)	Any AR after 1st dose (%)	56 (73.7)	2222 (63.7)	0.074 ^n^	295 (67.0)	1983 (63.5)	0.147
	Any AR after 2nd dose (%)	45 (59.2)	2335 (67.0)	0.156	291 (66.1)	2089 (66.9)	0.753
	Any mild AR after any dose (%)	29 (38.2)	1089 (31.2)	0.198	168 (38.2)	950 (30.4)	0.001
	Any mild AR after 1st dose (%)	21 (27.6)	667 (19.1)	0.063 ^n^	102 (23.2)	586 (18.8)	0.028
	Any mild AR after 2nd dose (%)	17 (22.4)	622 (17.8)	0.308	99 (22.5)	540 (17.3)	0.008
	Any severe AR after 2nd dose (%)	13 (17.1)	1036 (29.7)	0.017	108 (24.5)	941 (30.1)	0.016
Proportion of specific ARs (% of vaccinees with an AR)	**Fever after 2nd dose (%)**	**11 (14.5)**	**974 (27.9)**	**0.010**	114 (25.9)	871 (27.9)	0.380
	**Malaise after any dose (%)**	**7 (9.2)**	**622 (17.8)**	**0.050**	68 (15.5)	561 (18.0)	0.200
	**Malaise after 2nd dose (%)**	**4 (5.3)**	**517 (14.8)**	**0.020**	56 (12.7)	465 (14.9)	0.230
	**Severe malaise after any dose (%)**	**1 (1.2)**	**305 (8.8)**	**0.010**	32 (7.3)	274 (8.8)	0.290
	**Severe malaise after 2nd dose (%)**	**1 (1.2)**	**258 (7.4)**	**0.030**	25 (5.7)	234 (7.5)	0.170
	Severe arm pain after 2nd dose (%)	1 (1.3)	187 (5.4)	0.120	15 (3.4)	173 (5.5)	0.060 ^n^
	Chills after 2nd dose (%)	2 (2.6)	385 (11.0)	0.020	35 (8.0)	352 (11.3)	0.040
	**Myalgia after any dose (%)**	**5 (6.6)**	**709 (20.3)**	**0.003**	94 (21.4)	620 (19.9)	0.460
	**Myalgia after 2nd dose (%)**	**4 (5.3)**	**621 (17.8)**	**0.004**	79 (18.0)	546 (17.5)	0.810
	Severe myalgia after 2nd dose (%)	2 (2.6)	296 (8.5)	0.070 ^n^	29 (6.6)	269 (8.6)	0.150
	Headache after any dose (%)	11 (14.5)	967 (27.7)	0.010	118 (26.8)	860 (27.5)	0.750
	Headache after 2nd dose (%)	8 (10.5)	750 (21.5)	0.020	89 (20.2)	669 (21.4)	0.570
	Severe headache after any dose (%)	3 (3.9)	419 (12.0)	0.030	39 (8.9)	383 (12.3)	0.040
	Severe headache after 2nd dose (%)	2 (2.6)	331 (9.5)	0.040	28 (6.4)	305 (9.8)	0.020

NS: Non-significant; *p* ≤ 0.05: statistically significant; statistically significant results that are consistent in the three analyses (overall immunogenicity analysis) are shown in bold type; ^n^ AR for which a positive nominal trend was observed; * Median value of antibody titres (P25, P75) = 10.00 (10.00, 10.00) on a semiquantitative scale ranging from 0 to >10.

**Table 4 vaccines-12-00665-t004:** Results of sensitivity analysis 2.

Sensitivity Analysis 2							
N = 597		Antibody Positiveness		*p*-Value	Antibody Titres		*p*-Value
		Negative Antibody Titres (n = 273)	Positive Antibody Titres (n = 324)		Antibody Titres below Median * (n = 288)	Antibody Titres above Median * (n = 309)	
Intensity (maximum intensity grade; % vaccinees at each grade)	Grade 1 after 1st vaccine dose (%)	48 (17.6)	39 (12.0)	0.050 ^n^	50 (26.2)	37 (17.5)	0.072 ^n^
	Grade 2 after 1st vaccine dose (%)	79 (28.9)	94 (29.0)		81 (42.4)	92 (43.4)	
	Grade 3 after 1st vaccine dose (%)	53 (19.4)	90 (27.8)		60 (31.4)	83 (39.2)	
Intensity (mean, SD Likert score)	After 1st dose (mean, SD)	3.37 (2.99)	3.83 (3.21)	0.083 ^n^	5.28 (2.18)	5.65 (2.09)	0.083 ^n^
	After 2nd dose (mean, SD)	3.63 (3.31)	3.01 (3.23)	0.001	5.87 (2.13)	5.77 (2.02)	0.570
Proportion of ARs (% of vaccinees with)	Any AR after 2nd dose (%)	187 (68.5)	188 (58.0)	0.008	193 (67.0)	182 (58.9)	0.040
	Any moderate AR after 1st dose (%)	99 (36.3)	139 (42.9)	0.099	103 (35.8)	135 (43.7)	0.048
	Any moderate AR after 2nd dose (%)	126 (46.2)	124 (38.3)	0.052 ^n^	129 (44.8)	121 (39.2)	0.163
	Any severe AR after 1st dose (%)	53 (19.4)	90 (27.8)	0.017	60 (20.8)	83 (26.9)	0.085 ^n^
Proportion of specific ARs (% of vaccinees with)	Fatigue after 1st dose (%)	21 (7.7)	20 (6.2)	0.465	24 (8.3)	43 (13.9)	0.031
	**Fever after 1st dose (%)**	**16 (5.9)**	**73 (22.5)**	**<0.001**	**19 (6.6)**	**70 (22.7)**	**<0.001**
	**Severe fever after 1st dose (%)**	**0 (0.0)**	**11 (3.4)**	**0.002**	**1 (0.3)**	**10 (3.2)**	**0.009**
	**Malaise after any dose (%)**	**37 (13.6)**	**76 (23.5)**	**0.002**	**41 (14.2)**	**72 (23.3)**	**0.004**
	**Malaise after 1st dose (%)**	**5 (1.8)**	**36 (11.1)**	**<0.001**	**7 (2.4)**	**34 (11.0)**	**<0.001**
	**Severe malaise after any dose (%)**	**17 (6.2)**	**42 (13.0)**	**0.006**	**19 (6.6)**	**40 (12.9)**	**0.009**
	**Severe malaise after 1st dose (%)**	**3 (1.1)**	**18 (5.6)**	**0.003**	**5 (1.7)**	**16 (5.2)**	**0.023**
	Chills after 1st dose (%)	2 (0.7)	25 (7.7)	<0.001	4 (1.4)	23 (7.4)	0.004
	**Myalgia after 1st dose (%)**	**9 (3.3)**	**36 (11.1)**	**0.003**	**9 (3.1)**	**36 (11.7)**	**<0.001**
	**Severe myalgia after 1st dose (%)**	**3 (1.0)**	**18 (5.8)**	**0.003**	**3 (1.0)**	**18 (5.8)**	**0.002**
	Other musculoskeletal disorders after 1st dose (%)	2 (0.7)	9 (2.8)	0.064 ^n^	3 (1.0)	8 (2.6)	0.160
	Arthralgia after 1st dose (%)	0 (0.0)	10 (3.1)	0.034	1 (0.3)	9 (2.9)	0.015
	Severe arthralgia after 1st dose (%)	0 (0.0)	6 (1.9)	0.024	1 (0.3)	5 (1.6)	0.120
	Headache after 1st dose (%)	24 (8.8)	48 (14.8)	0.024	26 (9.0)	46 (14.9)	0.028
	Severe headache after 1st dose (%)	6 (2.2)	20 (6.2)	0.018	6 (2.1)	20 (6.5)	0.009

NS: Non-significant; *p* ≤ 0.05: statistically significant; statistically significant results that are consistent in the three analyses (overall immunogenicity analysis) are shown in bold type; ^n^ AR for which a positive nominal trend was observed; * Median value of antibody titres (P25, P75) = 2.53 (0.47, 10.00) on a semiquantitative scale ranging from 0 to >10.

## Data Availability

Complete data are available in the Appendix A. Raw data are available from the corresponding author upon reasonable request.

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
