# Peer review of "Association between Reactogenicity and Immunogenicity in a Vaccinated Cohort with Two mRNA SARS-CoV-2 Vaccines at a High-Complexity Reference Hospital: A Post Hoc Analysis on Immunology Aspects of a Prospective Cohort Study"

_vaccines, 2024, doi:10.3390/vaccines12060665_

Round 1

Reviewer 1 Report (Previous Reviewer 1)

Comments and Suggestions for Authors

In this revised manuscript, the authors have made some efforts to address the comments and suggestions provided in the first round of the review process and have achieved certain improvements in both data presentation and discussion. However, improvements are still needed for the manuscript to be potentially acceptable for publication.

My major concerns are listed here:

1.        Many of the specific reactogenicities claimed by the authors to be associated with immunogenicity overlap with the most common adverse reactions related to mRNA vaccines. How useful are these reactogenicities in facilitating the prediction of immunogenicity? Is it possible for the authors to test grouping these parameters as improved signature indicators of good antibody responses post-vaccination?

2.        Is there any possibility that the authors can rank the specific reactogenicities that are closely associated with immunogenicity in clinical settings?

Some minor comments:

1.        Figures 2-4 need to be simplified to emphasize the most significant results extracted from the corresponding tables.

2.        Add y-axis labels and indicate statistical significance on Figures 2-4.

Author Response

Reviewer 2 Report (Previous Reviewer 2)

Comments and Suggestions for Authors

The manuscript is still hard to read through even after revision. The manuscript is still too long and hard to follow. The manuscript can be further shortened by compacting and organizing the content. And the description is not clear. I cannot find Materials and Methods section 2.2.

Tables are still hard to be figured out. For example, the description of “Quantitative analysis” in Tables is not clear. Authors used the word, “Cutoff by the median”. But the meaning is not clear. Authors should describe “the median of what” in Tables. Authors describe “a positive nominal trend was observed” when p-value was between 0.05 and 0.10 in Tables. It is a strange description.

The usage of words is not constant. It makes hard for readers to read through the manuscript. For example, “structured questionnaire” is used in line 100. However, “structured interview” is used in line 618. “primary analysis” is used in line 224, but “main analysis” is used in Table 2 and other places.

There are remaining many points to be improved in the manuscript.

Comments on the Quality of English Language

There are many careless and grammatical errors though manuscript.
Authors should check English carefully before submission.
English should be brushed up thoroughly.

Author Response

Reviewer 3 Report (Previous Reviewer 3)

Comments and Suggestions for Authors

Dear Authors:

Second Round for the Review of the manuscript: “Association between reactogenicity and immunogenicity in a vaccinated cohort with two mRNA SARS-CoV-2 vaccines at high complexity reference hospital”

The work resubmitted presents now (although in supplementary materials) some information on cardiovascular (CV) events that were bluntly omitted in the previous version. Nevertheless, the authors still claim that the CV events are not clearly associated with the mRNA vaccines and conveniently omitted a systematic review on the topic that claims that: “CV events such as thrombosis, thrombocytopenia, stroke, and myocarditis frequently occur with the mRNA vaccines studied.” [1] (in reference to COVID-19 mRNA vaccines) . Moreover, the authors claimed that the association is low: “Likewise, a recent umbrella review of published systematic reviews and meta-analyses found an incidence of 0.89 to 2.36 cases of myocarditis per 100,000 doses of vaccine”, data that is based on a study that reviewed information about all COVID-19 vaccines (all vaccine platforms together) and not only mRNA vaccines [2]. Although this could be seen as an effort to justify previously neglected CV events, a worst effect is that this will mislead the reader to believe that mRNA vaccines causes just 0.89 to 2.36 cases of myocarditis per 100,000, that is not true. To use not pertinent information to inflate or bias a discussion, and for sure to confound readers is enough ground for rejection. 

For the mentioned reasons the manuscript is still rejected.

The reviewer

References:

[1] Yasmin F, Najeeb H, Naeem U, Moeed A, Atif AR, Asghar MS, Nimri N, Saleem M, Bandyopadhyay D, Krittanawong C, Fadelallah Eljack MM, Tahir MJ, Waqar F. Adverse events following COVID-19 mRNA vaccines: A systematic review of cardiovascular complication, thrombosis, and thrombocytopenia. Immun Inflamm Dis. 2023 Mar;11(3):e807. doi: 10.1002/iid3.807. PMID: 36988252; PMCID: PMC10022421.

[2] Bouchlarhem A, Boulouiz S, Bazid Z, Ismaili N, El Ouafi N. Is There a Causal Link Between Acute Myocarditis and COVID-19 Vaccination: An Umbrella Review of Published Systematic Reviews and Meta-Analyses. Clin Med Insights Cardiol. 2024 Jan 18;18:11795468231221406. doi: 10.1177/11795468231221406. PMID: 38249317; PMCID: PMC10798131.

Author Response

Reviewer 4 Report (Previous Reviewer 4)

Comments and Suggestions for Authors

This manuscript has been carefully and intensively revised. Only minor editing or formatting are needed.

Check lines 106, 174, and 290.

Round 2

Reviewer 2 Report (Previous Reviewer 2)

Comments and Suggestions for Authors

The manuscript was well improved after vast revision.

I just comment small points. Authors should check the manuscript again carefully.

Lines 89, 94. “7th” and “2nd” are not superscripted.

Lines 474, 475. “significant association between chills, headache, injection, site pain ,arm pain, and arthralgia”. Usually “between” is used for comparison of A and B. It is not clear that the sentence means association between what and what. Describe it clearly.

Line 478. “Speletas M et al.” reads “Speletas et al.”

Line 495. “less than for” reads “less than with”. Line 498. “SARS-CoV-2a” reads “SARS-CoV-2”.

Line 586. What does “RWE studies” stand for?

Line 611. “SmPC” should be spelled out. SmPC (summary of product characteristics) In relation to this, PRAC Meeting in references#37 and #38 should be spelled out. Description of date of the events are not necessary (lines 35, 37).

References. Check the format of References again. Some references show three authors’ names and “et al.” and some references show six authors’ names and “et al.” And some references show more than 6 authors’ names. Check them.

Author Response

This manuscript is a resubmission of an earlier submission. The following is a list of the peer review reports and author responses from that submission.

Round 1

Reviewer 1 Report

Comments and Suggestions for Authors

In this observational study utilizing real-world data, the authors conducted a comparative analysis across three distinct populations that received two mRNA SARS-CoV-2 vaccines. They illustrated a significant association between certain high-incidence adverse reactions (ARs), such as fever, malaise, and myalgia, and positive antibody response, rather than cellular immunity. Given the inconsistency within existing literature regarding this correlation between systemic and local ARs with immunoreactivity, the authors' findings among diverse vaccinated populations offer valuable insights in this field. I recommend considering potential acceptance, contingent on significant revisions in terms of data presentation.

My major comments are outlined below:

Enhance Figure 1 to improve clarity to facilitate better understanding of the data analysis flowchart in relation to other results sections, clarifying how exactly patient data are compared across groups.

Simplify most table results for enhanced readability, aiding readers in grasping novel discoveries. Key findings should be extracted and presented in figures to highlight their importance.

Incorporate correlation figures to demonstrate the close association between reactogenicity and immunogenicity, particularly concerning antibody immune response as opposed to cellular responses.

Present direct comparison results between Healthcare Professionals (HCP) and Solid-Organ Transplant Recipients (SOTR) to highlight any discrepancies in reactogenicity. Lower reactogenicity observed in SOTR compared to HCP would bolster the validity of the manuscript's conclusions.

Conduct a direct comparison of the two different mRNA vaccines to rule out the possibility of certain reactogenicities stemming from vaccine composition rather than immunogenicity.

Comments on the Quality of English Language

The quality of English writing is good; no major issues were found

Reviewer 2 Report

Comments and Suggestions for Authors

The manuscript shows that frequency of vaccine adverse reactions such as fever, malaise, and myalgia was associated with development and quantity of antibody titers elicited by SARS-CoV-2 mRNA vaccines. In addition, the association between reactogenicity and cellular immunogenicity was also assessed in the subpopulation of the subjects. The data shown in the manuscript is important. However, the manuscript should be improved. First of all, the manuscript should be shortened.

Table 2 is too large and too complex. It is hard to read and needs to be revised to be more reader-friendly. The table should be separated to several tables. The description of “Median<>Q2” is strange.

The supplementary figures are also too large and hard to read. The format of the figures is partly broken and it should be improved.

Reviewer 3 Report

Comments and Suggestions for Authors

Review of the manuscript: “Association between reactogenicity and immunogenicity in a vaccinated cohort with two mRNA SARS-CoV-2 vaccines at high complexity reference hospital”

The work present an observational data analysis based on patient electronic records stored at Hospital Clínic of Barcelona, Spain. The paper used information that measured several vaccine related adverse effects but mainly focused on few (i.e.: myalgia, fever and malaise), claiming a correlation of vaccine immunogenicity and reactogenicity. 

The work has several limitations and biases that impede its publication as valuable scientific information. The data about adverse events is apparently not showing any cardiac complication, or this kind of questions were deliberately not included in the questionnaire for adverse events (obtained by telephone call), which is worst. Cardiac complications are an important issue in mRNA vaccines, which apparently arise from its systemic spike protein dissemination after covid vaccination. One of the problems of this cardiac complications caused by mRNA vaccination is actually the “underreporting”. I have serious concerns letting authors publishing data that will increase the underreport of an important adverse event, which also has a well described mechanism for its associated pathology. 

Pitifully, for this reason (and other minor reasons not mentioned here) the manuscript is rejected.

The reviewer.

Reviewer 4 Report

Comments and Suggestions for Authors

1.       As this report is based on the authors’ previous VigilVacCOVID study, a brief background on the published VigilVacCOVID study should be included in the Introduction.

2.       In the Conclusion section, the authors provided a piece of additional background information and some discussion (We consider that our results can be impactful to advance our understanding…); but a conclusion is missing in the Conclusion.

3.       The Abstract and the Introduction sections are fine; Other sections may need some editing (checking: lines 224-227; Table 2, line 452-453; line 555 and other places).

4.       Some wordings and expressions may be improved. Taking figure 1 as an example, in a box of the figure, “Exclusion of vaccinees with immunogenicity assessments prior to vaccination: 225 participants”. This may lead to a misunderstanding as that the 225 participants are the number excluded. In fact, the number is AFTER exclusion of vaccinees with immunogenicity ….

Also, in a box in figure 1, “Participants with information on AR and immunogenicity and vaccinated with BNT162b2” To make it easier to follow, how about we change it to, “BNT162b2 vaccinees with information on AR and immunogenicity” or “BNT162b2-vaccinated participants with information …”  These are just minor suggestions. 

Comments on the Quality of English Language

To improve the paper, it will be better to have an editor to go through the paper, especially the Results, Discussion and Conclusion sections.
